# Realization of Unity Power Factor Wireless Power Transfer System under Subnormal Operation Conditions

**Liyong Zhang** [1], **Pengyu Zhang** [1], **Wenwu Li** [1,*] **and Zhonghao He** [2]

1   College of Electrical Engineering and New Energy, Three Gorges University, Yichang 443002, China; 987yyy@ctgu.edu.cn (L.Z.); yuyuyu9797@ctgu.edu.cn (P.Z.)
2   Hubei Electric Power Corporation, State Grid Corporation of China, Yichang 443002, China; zhonghaohe@ctgu.edu.cn
*   Correspondence: liww@ctgu.edu.cn

**Abstract:** The power factor of wireless power transfer system, determined by its compensation network part, is easily affected by parameter detuning, coil misalignment, and load variation. In this paper, a mathematical model for the compensation network part is established. Theoretical analysis shows that the inverter part can be considered as a negative resistor by deducing the inherent static-state frequency solution of the compensation network part. Therefore, the unity power factor wireless power transfer system can be maintained under any possible operation conditions by tracking the inherent static-state frequency solution. More importantly, no digital controller or parameter identification or information interactions between the primary and secondary coils are needed during the tracking process. Compared with previous unity power factor realization methods, the proposed tracking strategy has the advantages of fewer sampling variables, a faster response time, and a simpler regulation process. Finally, an experimental platform is built to test the practical performance of the proposed tracking strategy under many subnormal operation conditions. Our experimental results show that approximate unity power factor can be realized at 10–15 cm coil misalignment distance and 30–90 Ω load variation range.

**Keywords:** wireless power transfer; parameter detuning; subnormal operation; unity power factor; soft switching

## 1. Introduction

As one of the special transmission technologies for medium- and short-distance electric energy, wireless power transfer (WPT) technology has attracted more and more attention due to its unique advantages with respect to safety, convenience, and other advantages. Its applications mainly include low-power demand scenarios, such as those involving electronic equipment [1], smart home devices [2], and medical devices [3], and high-power energy transmission scenarios, such as those involving electric vehicles and industrial robots [4,5].

For a WPT system, various unavoidable factors, such as time-varying loads, uncertain coil distances, parameter detuning, and interoperability scenarios, can introduce deviations from normal operating conditions [6,7]. Moreover, the ruination of the normal operation state brings undesired consequences, such as a decrease in power factor (PF), a reduction in transmission efficiency, a loss of soft switching, or an increase in component stress. Among them, PF is closely correlated with transmission efficiency, component stress, and soft switching. A larger PF helps to improve the transmission efficiency, realize soft switching, and reduce component stress. Normally, the compensation network parameters determine the power factor of the WPT system, and the networks are designed as unity PF systems in order to fully compensate for reactive power and achieve higher transmission efficiency [8]. Nevertheless, parameter variation can still lead to a reduction in the inverter's power factor. Therefore, the realization of a unity PF WPT system is particularly important.

Recently, many researchers have investigated how to obtain the unity PF WPT system [9–17]. The parameters in the compensation network part, are well designed under fixed operation frequency to realize the unity PF of the inverter [9]; however, this method works well for scenarios where only load or mutual inductance vary. Variation in the network parameters will still reduce the PF of the inverter. By comparison, the introduction of a switched controlled capacitor (SCC) structure can guarantee the realization of unity PF even if parameters detuning occurs [11]. However, the need for a large amount of sampling information during the feedback circuit makes this hard to execute. DC current controlled variable inductance structure can alleviate control complexity while compensating for parameter detuning [13]. However, the additional power source and conversion circuit still prevents it from further development. In [15], a self-adaptive mistuning correction circuit was proposed to ensure that the WPT system consistently operates in a resonant state. No closed-loop regulation is needed with this method, and the proposed method can self-adaptively eliminate either undesired capacitive or inductive reactance with two additional switches and an energy-storage capacitor. However, the addition of an extra self-adaptive mistuning correction circuit will reduce the power density and transmission efficiency of the whole WPT system.

In this paper, a two-coil WPT system with a series-none (SN) topology is adopted. The operation mechanism for the unity PF and static-state solution in the compensation network part is investigated. Then, the unity PF system can be consistently ensured by tracking the static-state solution under any network parameter. Furthermore, a tracking strategy can be proposed accordingly to ensure the unity PF of the whole WPT system under any subnormal operation scenario. Compared with the systems that have been described in similar studies, the proposed WPT system has a faster response time, no additional components, fewer sampling signals, and a simpler execution process.

## 2. Theoretical Analysis

### 2.1. Static-State Solution Derivation for SN Compensation

The compensation capacitor in the secondary side can be eliminated under a strong coupling condition [18,19], resulting in a higher power density, a smaller power loss, and a lower system cost in the receiver part, which is very suitable for applications with strict requirements regarding the volume and weight of the receiver device. A compensation capacitor was connected with a transferring coil in series to decrease the injection of reactive power [20]. Therefore, series-none (SN) compensation was adopted to ensure the establishment of a WPT system that can operate under strong coupling conditions.

The equivalent circuit of the proposed WPT system with a SN compensation is depicted in Figure 1, where $v_p$ and $v_o$ represent the input and output ac voltage, respectively; $i_p$ and $i_s$ are the input and output ac current, respectively; $L_p$ and $L_s$ are the self-inductance of the transmitting coil and receiving coil, respectively; $M$ is the mutual inductance between the transmitting and receiving coils, which determines the power transferred to the secondary side; $C_p$ is the primary compensation capacitor; $R_p$ and $R_s$ are the branch internal resistance of transmitter part and receiver part, respectively; and $R_L$ is the equivalent ac load.

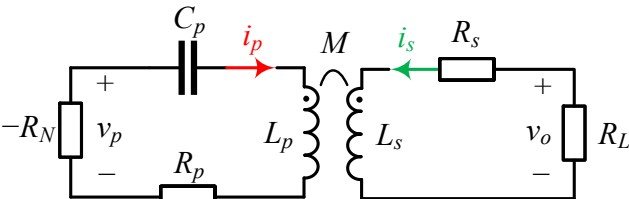

**Figure 1.** The equivalent circuit of the proposed S/N compensation network.

Assuming that the unity PF of the inverter is realized, which means the zero phase angle (ZPA) of $v_p$ and $i_p$ is achieved, the input impedance can be seen as a purely resistive component (expressed as $R_N$ in Figure 1). In other words, $-R_N$ is a negative resistance

that provides active power to the secondary load $R_L$. For the convenience of analysis, the fundamental components of the adopted voltages and currents are only considered in the following analysis process. Thus, $V_p$, $V_o$, $I_p$, and $I_s$ are used to replace $v_p$, $v_o$, $i_p$, and $i_s$, respectively. Then, according to circuit theory, the voltage equations of the transmitting and receiving circuits in the steady state can be described as

$$\begin{bmatrix} R_{N1} + j(\omega L_p - \frac{1}{\omega C_p}) & j\omega M \\ j\omega M & R_{L1} + j\omega L_s \end{bmatrix} \begin{bmatrix} I_p \\ I_s \end{bmatrix} = \begin{bmatrix} 0 \\ 0 \end{bmatrix} \tag{1}$$

where $R_{N1}$ and $R_{L1}$ are defined as

$$R_{N1} = -R_N + R_p \tag{2}$$

$$R_{L1} = R_L + R_s \tag{3}$$

Normally, if the static-state solution of the SN topology exists, the characteristic equation of the above coefficient matrix, as is shown in (4), should be solvable as

$$[R_{N1} + j(\omega L_p - \frac{1}{\omega C_p})](R_{L1} + j\omega L_s) + \omega^2 M^2 = 0 \tag{4}$$

Furthermore, to simplify the calculation process to solve $\omega$ under steady state, the real and imaginary part of (4) can be separated, respectively, as follows:
Real part:

$$R_{N1} R_{L1} + \omega^2 M^2 - (\omega L_p - \frac{1}{\omega C_p})\omega L_s = 0 \tag{5}$$

Imaginary part:

$$R_{L1}(\omega L_p - \frac{1}{\omega C_p}) + \omega L_s R_{N1} = 0 \tag{6}$$

There are two unknowns that need to be solved: $R_{N1}$ and $\omega$. Hence, by combining Equations (5) and (6), $R_{N1}$ is eliminated first to obtain a quadratic equation regarding $\omega^2$, which can be expressed as

$$\left(L_p C_p L_s^2 - M^2 L_s C_p\right)\omega^4 + \left(L_p C_p R_{L1}^2 - L_s^2\right)\omega^2 - R_{L1}^2 = 0 \tag{7}$$

From (7), when $L_s$, $L_p$, and $C_p$ are fixed to ensure the existence of solution $\omega_s$, the relationship between coupling coefficient $k$ and load $R_{L1}$ under the given network parameters should satisfy the following condition:

$$k \le \frac{L_p C_p R_{N1}^2 + L_s^2}{2 R_{L1} L_s \sqrt{L_p C_p}} \tag{8}$$

In this case, static-state solution $\omega_s$ serves to realize the $R_N$ operation, and it can be derived as

$$\omega_s = \sqrt{\frac{-\lambda_2 + \sqrt{\lambda_2^2 + 4\lambda_1 R_{L1}^2}}{2\lambda_1}} \tag{9}$$

where $\lambda_1$ and $\lambda_2$ are the coefficients of $\omega^4$ and $\omega^2$ in Equation (7), respectively.

From (8), provided that coupling coefficient $k$ is smaller than the critical $k'$ value under the given $R_{L1}$ and $C_p$ values, the switching frequency needed to make the input voltage and current realize the ZPA can be calculated from (9). The critical $k'$ value curve against load $R_L$ at different $C_p$ is shown in Figure 2. It can be observed that the critical $k'$ value is always greater than 1 at given $R_L$ variation ranges and $C_p$ values. In addition, in practice, the coupling coefficient is less than 1. Therefore, Equation (9) is solvable under any normal

operation condition, which also means that the unity PF operation can be realized amidst an extraordinarily wide load variation range and coupling coefficient.

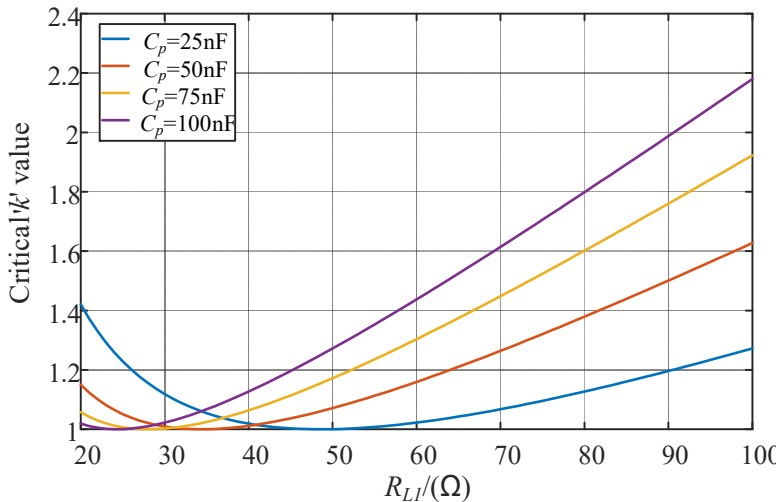

**Figure 2.** The critical $k'$ value curve against load $R_{L1}$ at different $C_p$.

Furthermore, the switching frequency curves required to realize unity PF operation against $C_p$ under different $M$ and $R_{L1}$ are depicted in Figure 3. The static-state $f_s$ at different $M$ and $R_{L1}$ keep the same variation trend. With an increase in the primary capacitor $C_p$ value, the static-state $f_s$ will decrease. With a larger $C_p$ value, the capacitive reactance at present switching frequency will decrease. Hence, the switching frequency should decrease accordingly to reduce the inductive reactance at a fixed $L_p$ and $L_s$ combination. In addition, the variation rate of static-state $f_s$ will drop slowly as $C_p$ increases. This occurs because the capacitive reactance becomes less sensitive to $C_p$ variation when the $C_p$ value is too large. Additionally, the frequency curves at different $R_L$ and $M$ are very close together, which means the operation frequency for input ZPA is not sensitive to the variation of $R_L$ and $M$. Hence, the input robustness of the whole WPT system can be greatly enhanced.

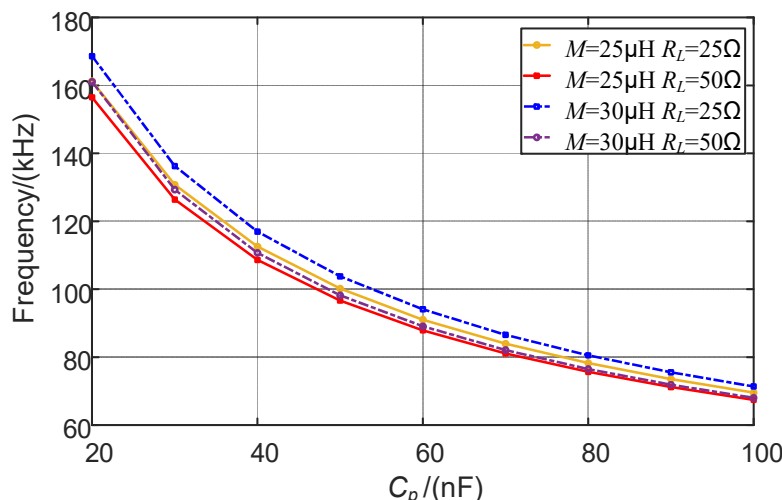

**Figure 3.** Switching frequency curves required to realize ZPA operation against $C_p$ under different $M$ and $R_L$.

### 2.2. Output Power and Coupling Efficiency Analysis of SN Compensation

According to the results of the above analysis, the operation angular frequency, eliminating the reactive power injection in the primary inverter part, can be deduced by (9) under any given $C_p$, $M$, and $R_{L1}$. In this case, through substituting (9) into (6), the equivalent negative resistance $R_N$ can be calculated as

$$R_N = \frac{\omega_s^2 M^2 R_{L1}}{\omega_s^2 L_s^2 + R_{L1}^2} + R_p \tag{10}$$

Assuming the input voltage amplitude is fixed as $V_p$, the input power $P_{in}$ can be expressed as

$$P_{in} = \frac{V_p^2}{R_N} = \frac{V_p^2(\omega_s^2 L_s^2 + R_{L1}^2)}{\omega_s^2 M^2 R_{L1} + (\omega_s^2 L_s^2 + R_{L1}^2)R_p} \tag{11}$$

Moreover, the coupling efficiency from the input port to ac load $R_L$ can be expressed as

$$\eta_{coil} = \frac{R_N - R_p}{R_N}\frac{R_L}{R_s + R_L} \tag{12}$$

Finally, the power transferred to $R_L$ can be calculated as

$$P_o = P_{in}\frac{R_N - R_p}{R_N}\frac{R_L}{R_s + R_L} \tag{13}$$

From (13), the output power is decided by $\omega_s$, $R_L$, and $M$ when the coupling coils are given. Hence, the power fluctuation caused by the variation of $R_L$ and $M$ can be compensated by adjusting the operation frequency $\omega_s$. Since $\omega_s$ is correlated with primary compensation capacitor $C_p$ from (9), the constant output power can be realized by regulating the $C_p$ value for variable $R_L$ and $M$. Assuming $V_p$ is 25 V, the output power variation curves against $C_p$ under different $R_L$ and $M$ are shown in Figure 4. It can be concluded from Figure 4 that the output power is proportional to the variation in $C_p$, making it possible to realize output power regulation. In addition, the adjustable output power range is relatively wide. For example, the minimum variation range of the output power is 107.8–174.6 W, whereas the variation range of the primary compensation capacitor $C_p$ is 20–100 nF, which occurs at $R_L = 25\ \Omega$ and $M = 25\ \mu\text{H}$. When $R_L$ is 50 $\Omega$ and $M$ is 25 $\mu\text{H}$, the output power varies from 117 to 323.5 W under the same $C_p$ variation range.

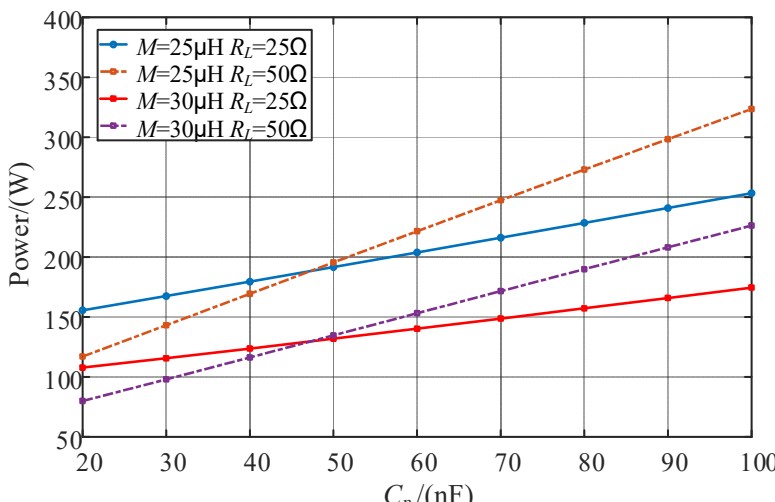

**Figure 4.** The output power curves against $C_p$ under different $R_L$ and $M$.

Owing to the purely resistive input impedance, no additional reactive power is introduced into the system. As a result, the input current can be reduced, generating a lower

degree of power loss throughout the whole compensation network. Then, the coupling efficiency can be increased. The coupling efficiency curves against primary capacitor $C_p$ with different $R_L$ and $M$ are shown in Figure 5. From this figure, it can be observed that the coupling efficiency can maintain a higher level during the $C_p$ variation range. According to Equation (9), a larger $C_p$ value results in a lower operation angular frequency $\omega_s$ under fixed $R_L$ and $M$ values. Further, the value of equivalent negative resistance $R_N$ in the input port will reduce accordingly. Therefore, the proportion of primary power loss in input power will increase due to the constant loop internal resistance in the primary side. Hence, the coupling efficiency will decrease as the $C_p$ increases. Additionally, with a larger $R_L$ and smaller $M$ value, the downward trend will be more obvious. This is because $R_N$ is inversely proportional to $R_L$ and directly proportional to $M$.

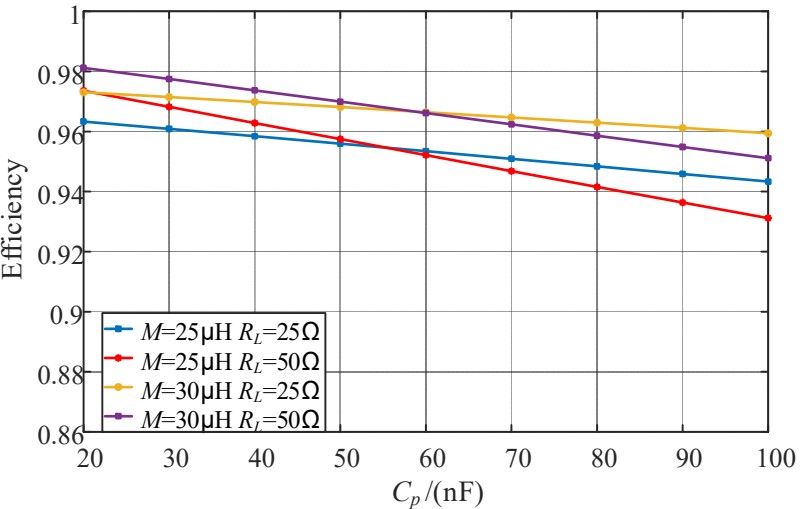

**Figure 5.** The coupling efficiency curves against $C_p$ with different $R_L$ and $M$.

## 3. Control Strategy

Based on the results of the above analysis, a control schematic of the proposed two-coil WPT system with a full bridge inverter is shown in Figure 6. The structure of a two-coil WPT system generally includes a high-frequency inverter, a control circuit, a resonant circuit, a high-frequency rectifier, and dc load $R_B$. Among them, the high-frequency inverter and control circuit at the transmitting side form a gain circuit, while the high-frequency rectifier and load at the receiving side form a loss circuit. The full bridge inverter converts the dc voltage into a high-frequency ac voltage source. In addition, only the resonant current in the primary side needs to be utilized as a feedback signal; the output voltage of the inverter can be to controlled and synchronized with the output current. Therefore, the full bridge inverter is equivalent to a negative resistor to provide the energy. As a result, the unity power factor of the inverter output can be guaranteed.

In addition, the control circuit used to realize the synchronization between inverter output voltage and the current is shown in Figure 7. The control circuit consists of a current detection circuit, a phase-shifting circuit, a comparison circuit, and a gate drive circuit. Its working principle uses current sensing to sample the resonant current signal at the transmitting side and convert it into an AC voltage signal. Then, the voltage signal is amplified by a differential amplifier and sent to a comparator through a phase-shifting circuit to detect the zero crossing of the signal. The output signal of the comparator is connected to the gate drive circuit to drive the switch transistor in the inverter. Through this self-regulating control circuit, the system can automatically track the working frequency to ensure synchronization between the inverter output voltage and current.

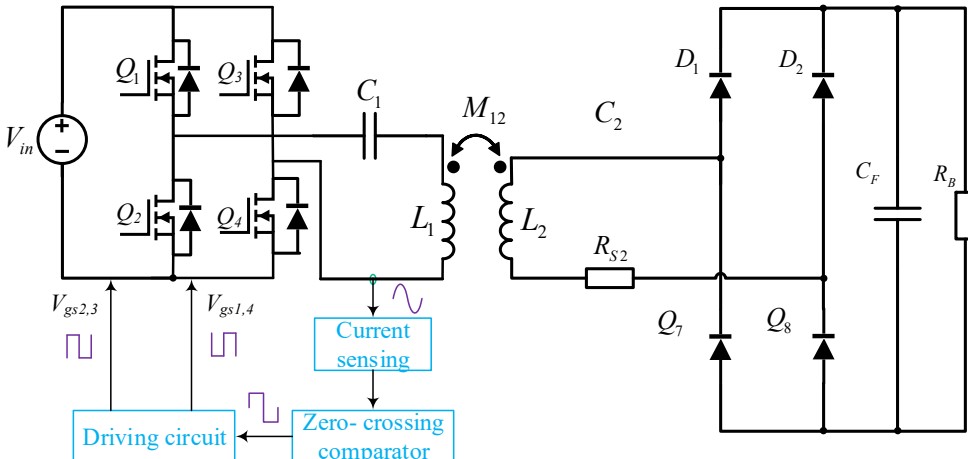

**Figure 6.** Control schematic of the proposed two-coil WPT system using a full bridge inverter.

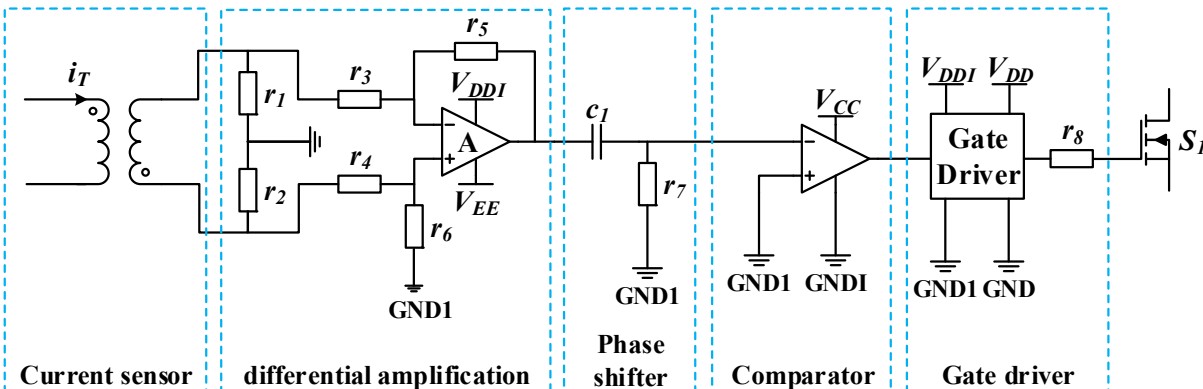

**Figure 7.** Control flowchart showing the realization of the synchronization between the inverter output voltage and current.

## 4. Experimental Performance

### 4.1. Experimental Platform

To verify the feasibility of our theoretical analysis, an experimental platform based on the proposed circuit structure was built, and it is shown in Figure 8. The component parameters in Table 1 were used to achieve the input requirements. It is important to note that the experimental setup was designed to confirm the practicability of the proposed circuit topology and control strategy. The whole experimental prototype can be adjusted properly when adopted to different applications.

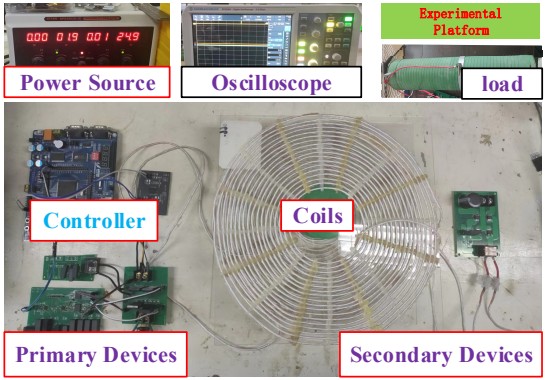

**Figure 8.** The experimental platform for the main circuit part.

**Table 1.** Specific components of the system.

| Parameters | $L_p$ | $C_p$ | $L_s$ | $R_p$ | $R_s$ |
|:---:|:---:|:---:|:---:|:---:|:---:|
| **Value** | 158.4 μH | 25 nF | 156.4 μH | 0.125 Ω | 0.120 Ω |

As for the form of the experimental platform, Infineon IRFP4227 was chosen to build the main high-frequency inverter part due to its low conduction resistance, and SI8233BD was selected as the gate driver to drive the MOSFET. A high-frequency current sensor, CU8966-ALC, converted the current from the primary resonant loop into an AC voltage signal. The full bridge rectifier on the secondary side is composed of four Schottky diodes (PSM20U200GS). The resonant capacitor uses a polypropylene film capacitor because of its stability under high-frequency conditions. At the same time, multiple nominal capacitors are connected in parallel, thus reducing the equivalent series resistance in practical applications, and the capacitor value error between the designed and measured resonant capacitor values can be reduced to almost zero.

In addition, the measured mutual inductance values against the offset distance at different directions are depicted in Figure 9. Assuming the rated transmission distance between the transferring and receiving coils is 10 cm in the z direction, then the mismatch between the transferring and receiving coils in the x, y, or z directions can all be explained by varying the *M* value.

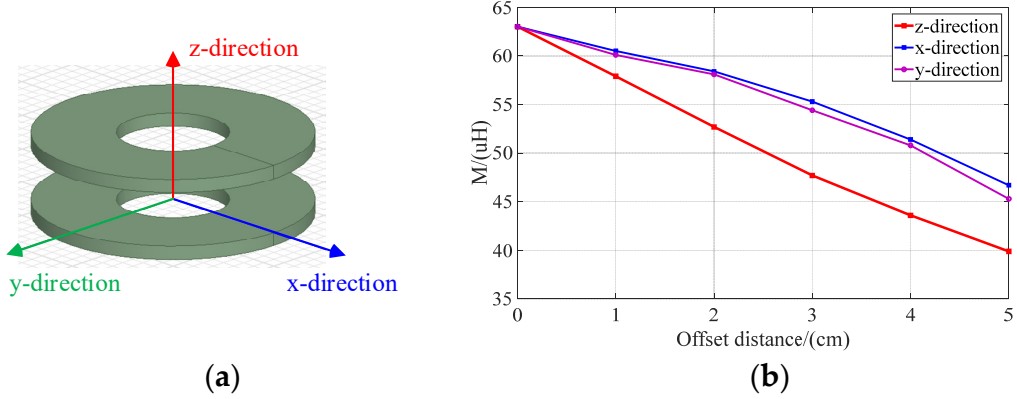

(**a**)　　　　　　　　　　　　　　　　　　　　　　　　(**b**)

**Figure 9.** Adopted coupling coils. (**a**) Coil structure. (**b**) Measured *M* value against offset distance.

*4.2. Experimental Results*

**(a) Unity power factor realization when $R_\mathrm{B}$ varies**

For a WPT system with a SN topology, input impedance will be affected by the variable load, and the power factor will change accordingly. By adopting the static-state frequency tracking strategy described in this paper, the imaginary part of the input impedance can be eliminated. Figure 10a,b show the input and output waveforms in 40 Ω and 60 Ω under fixed mutual inductance *M* and self-inductance $L_p$, respectively. By observing the number of grids in one period, the static-state resonant frequency can be measured as 92.3 kHz and 91.8 kHz, respectively. The phase angle error between input voltage and current is almost zero, in other words, the approximate unity power factor of the inverter part can be realized when load varies, which proves the practicality of the proposed tracking strategy.

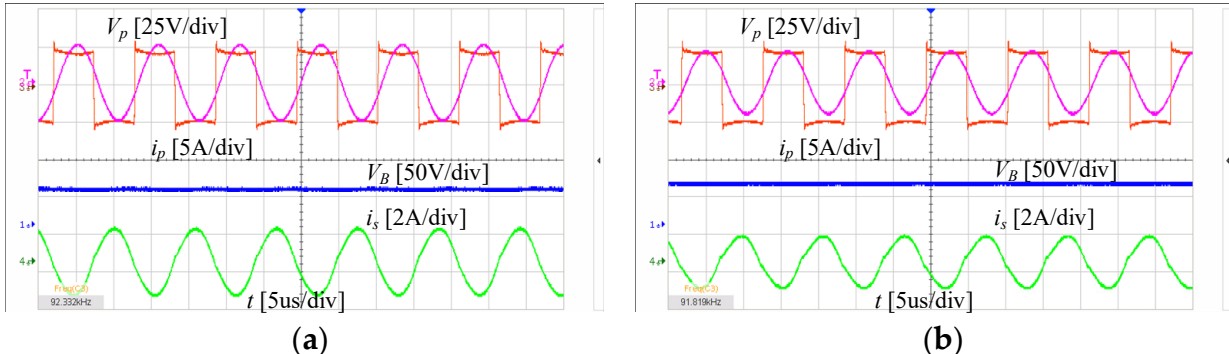

**Figure 10.** The waveforms under fixed mutual inductance $M$ and self-inductance $L_p$. (**a**) $R_B = 40 \ \Omega$; (**b**) $R_B = 60 \ \Omega$.

**(b) Unity power factor realization when $M$ varies**

Similarly, mutual inductance variation caused by coil misalignment will also lead to deviations in the input impedance for the SN topology. Unfortunately, coil misalignment is unavoidable. The voltage and current waveforms that appear when coil misalignment occurs are shown in Figures 11a and 11b, respectively. Based on Figure 11, it is clear that when $M$ changes from 60 μH (transmission distance is 10 cm) to 40 μH (transmission distance is 15 cm), the resonant frequency is regulated and decreases from 92.1 kHz to 89.6 kHz to maintain the purely resistive input characteristic. The unity power factor of the inverter part can also be achieved for any possible $M$ value. Therefore, the normal operation state can be guaranteed when coil misalignment occurs, avoiding potential safety issues.

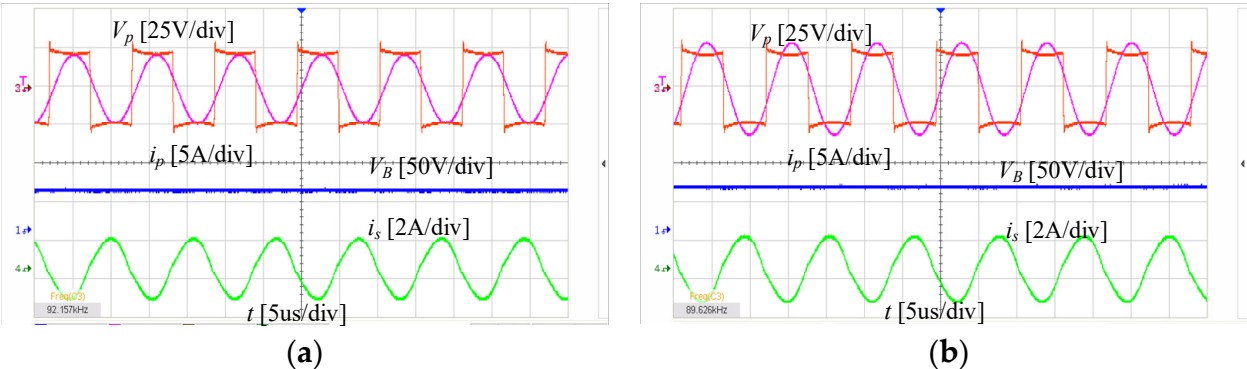

**Figure 11.** The waveforms under fixed load $R_B$ and self-inductance $L_p$. (**a**) $M = 40$ μH. (**b**) $M = 60$ μH.

**(c) Unity power factor realization when circuit parameters vary**

Circuit parameters are highly correlated with the input impedance characteristics. The addition of a magnitude core component in the coupling coils will cause self-inductance variation when coil misalignment occurs. Therefore, the primary self-inductance $L_p$ is chosen to verify the regulation ability of the adopted tracking strategy for parameter detuning. The waveforms of the input current $i_p$, input voltage $v_p$, output current $I_B$, and output voltage $V_B$ at different $L_p$ values are shown in Figure 12. Similarly, the operation frequency can be regulated automatically to maintain the unity power factor when $L_p$ varies. In addition, the response time does not exceed ten operation periods, allowing for fast, dynamic applications.

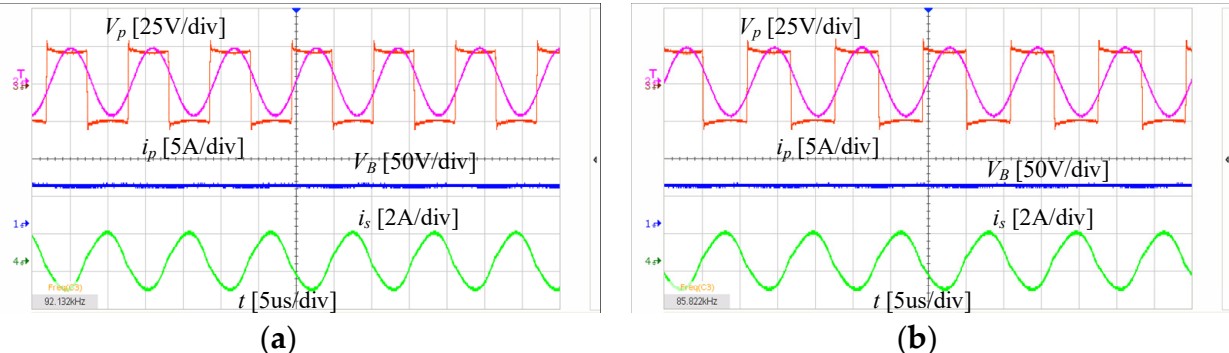

**Figure 12.** The waveforms under fixed load $R_B$ and mutual inductance $M$. (**a**) $L_p$ = 158.4 μH. (**b**) $L_p$ = 146.5 μH.

The efficiency curves of the whole WPT system against $R_B$ at different mutual inductance conditions are depicted in Figure 13. When $M$ is 60 μH, the transmission efficiency increases to 85.7% with the initial increase in load and subsequently drops slowly. Additionally, the lowest efficiency during the charging process is 78.2%, and this value was recorded when the mutual inductance was smaller. Moreover, it is also far higher than that reported in similar studies at the same power level.

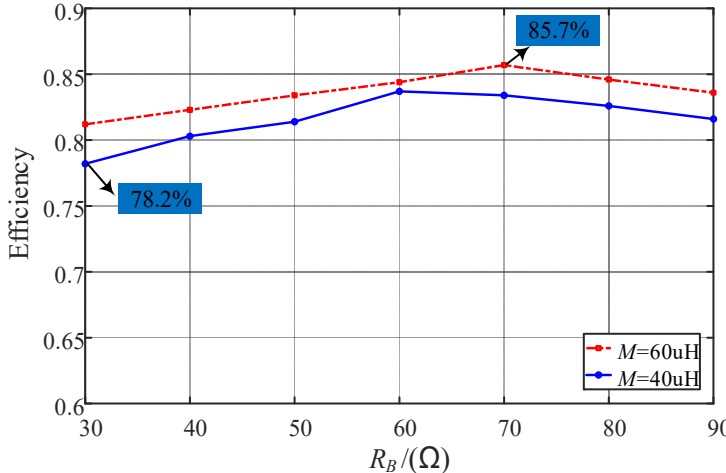

**Figure 13.** The measured efficiency curve against load under different mutual inductance conditions.

A performance comparison with systems described in similar studies is given in Table 2. Comparing our proposed system to that of similar studies, in our system, no additional component is needed, and the power density can be enhanced greatly. In addition, apart from the variation in load and coil misalignment, the reduction in unity power factor caused by the deviation of network parameters can also be avoided by the proposed system. However, the transmission efficiency of the proposed WPT system is a little low, though it can be improved by adopting a more suitable compensation network or through other efficiency improvement methods (for example, the coupler design method) [21–23].

**Table 2.** Performance comparison with systems described in similar studies.

|  | Additional Inductor | Additional Capacitor | Additional Switch | Power/(W) | Efficiency | Application |
|---|---|---|---|---|---|---|
| Proposed | 0 | 0 | 0 | 50 | 85.7% | $R_B$, $M$ and parameters vary |
| [9] | 1 | 2 | 1 | 2000 | 95% | $R_B$ varies |
| [11] | 0 | 2 | 2 | 800 | 96.8 | $R_B$ and $M$ vary |
| [15] | 1 | 2 | 2 | 59.6 | 88.6% | $R_B$ and $M$ vary |

## 5. Conclusions

A two-coil WPT system utilizing a SN compensation network has been proposed. Unity power factor is necessary to eliminate the reactive power and improve the transmission efficiency of the WPT system. Theoretical analysis shows that unity power factor can be maintained by tracking the inherent frequency characteristics under any potential fortuitous occasions. Then, a tracking strategy by detecting the primary resonant current can be proposed to guarantee the realization of unity power factor. The proposed tracking strategy performs well under parameter detuning, coil misalignment and load variation scenarios and has strong robustness. Finally, an experimental platform was constructed to verify the practical performance of the proposed WPT system. Through using our proposed system, power factor can be maintained at a relatively high level (above 0.95) under different operation conditions. Compared to the approaches described in similar studies, our proposed tracking strategy has a faster response time and simpler execution process.

**Author Contributions:** Conceptualization, Z.H.; methodology, L.Z.; software, W.L.; validation, L.Z. and W.L.; formal analysis, L.Z.; data curation, L.Z.; writing—original draft preparation, W.L.; writing—review and editing, P.Z. All authors have read and agreed to the published version of the manuscript.

**Funding:** This research received no external funding.

**Data Availability Statement:** All of the data supporting the reported results have been included in this paper.

**Acknowledgments:** We would like to thank the reviewers for their valuable comments. Some of the revisions have been taken directly from the suggestions of the reviewers.

**Conflicts of Interest:** The authors declare no conflict of interest.

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
