# Peer review of "Realization of Unity Power Factor Wireless Power Transfer System under Subnormal Operation Conditions"

_electronics, doi:10.3390/electronics12194009_

Round 1

Reviewer 1 Report (Previous Reviewer 1)

This new version of the manuscript has been improved compared to the previous version. It requires some minor revisions to make this article ready for publication.

1-      The exact value of the proposed tracking strategy performance improvement should be mentioned in the abstract.

2-      What is the relationship between mutual inductance variation due to coils misalignment and the input impedance deviation in the SN topology, and how does this affect the resonance frequency and power factor of the inverter part in the presence of misalignment?

3-      Mention the role and function of the full bridge inverter in the proposed two-coil wireless power transfer (WPT) system

4-      Improve the quality of the text inside figure 7.

5-      The font of the heading of the third section is different.

The writing in this manuscript is clear and informative.

Author Response

Reviewer 2 Report (Previous Reviewer 2)

This is a resubmitted manuscript, and I have made comments on this manuscript four months ago. In the email, I have already found the response towards my previous comments, and I found all the issues I proposed have been well addressed.

This manuscript focuses on transmission efficiency and the drop effect of the power factor. It should be noted that eddy current loss may reduce the efficiency, new magnetic core materials are needed to construct magnetic couplers to improve the efficiency of wireless charging systems. Some latest examples should be supplied at the beginning of the Introduction Section (10.3390/en15228363; 10.1109/TIE.2020.3016259).

Author Response

Reviewer 3 Report (Previous Reviewer 3)

1. In Fig. 13, the efficiency unit is missing.

2. In the Conclusion, it would be good to write a few specific conclusions that may be useful for designers of WPT systems.

3. What are the further directions of development of the proposed solution?

4. There are a few minor things to improve in the article related to the formatting of the article.

Moderate editing of English language required.

Author Response

This manuscript is a resubmission of an earlier submission. The following is a list of the peer review reports and author responses from that submission.

Round 1

Reviewer 1 Report

About the article titled” Realization of Unity Power Factor Wireless Power Transfer 2 System Under Subnormal Operation Occasions”, which address a mathematical solution for misalignment PF tolerance without digital feedback between TX/RX system I think the paper is addressing quite an important problem of WPT systems and its method is quite interesting. Therefore, I only address some points for the authors just for better visualization and improvement the quality of the paper as below:

1.       Page 2 line 57: what is S/N topology? better for addressing it serial-none here first

2.       I suggest adding a comparison table at the end of the measurement parts to compare your work in case of efficiency, the power delivered to load, working frequency, size of coils, transfer distance and appropriate applications to conclude the work in a much more effective way. And compare your work with the pros and cons of similar papers such as (DOI: 10.3390/en15228643)

3.       I couldn’t find the transmitting distance.

4.       You have mentioned that the work is appropriate for misalignment tolerance. How did you show this? Please depicted some figures about this. For example, x-y or h (rotation) mismatch between TX/RX coils and see the PF decrement plot.

5.       What is the drawback of your work?

6.       85 % efficiency is a high amount. Can you give more details of measurement efficiency?

Minor editing of English language required.

Reviewer 2 Report

1. I suggest that the authors increase the citation of relevant papers published in Electronics recently. The authors may consider using zero phase angle (ZPA) or zero voltage switching (ZVS) as search criteria, such as https://doi.org/10.3390/electronics12020463

2. The method proposed in this paper for tracking the inherent static-state frequency solution appears to be better suited for promoting the interoperability of wireless charging devices. The interoperability of wireless charging systems refers to the ability of output performance to meet specified indicators when different transmitter and receiver devices are matched. I suggest that the authors revise line 32 to emphasize the significance of their approach in the following way:

For a WPT system, various unavoidable factors such as time-varying load, uncertain coil distance, parameter detaining and interoperability scenarios, can introduce deviations from normal operating conditions.[https://doi.org/10.3390/en16041653]

3. In relation to the matter of power factor, the authors should begin with a discussion of the fundamental topology. For instance:

The compensation topology parameters determine the power factor of wireless charging systems. Designers often aim to design compensation networks as unity PF systems in order to fully compensate for reactive power and achieve higher transmission efficiency [10.3390/en16073084]. Nevertheless, parameter variations can still result in a reduction of the inverter's power factor.

4. It is stated on Line 183 that a half-bridge inverter is used as shown in Fig. 6, whereas the inverter depicted in the figure is actually a full-bridge inverter.

5. The self-oscillating control inverter mentioned on Line 196 does not correspond to the inverter shown in Fig. 6.

6. Table 1 lacks standardization in its formatting.

7. Additionally, the term 'RB' is used in the experimental section without prior definition, and the X-axis of Fig. 13 is labelled as 'RL.'

8. The font used in the figures should be standardized as either Times New Roman or Palatino Linotype.

The article presents an approach for maintaining a unit power factor in wireless charging systems, by tracking the inherent static-state frequency solution. The proposed method is considered highly applicable. However, the content should be further improved and the English should be further modified to be more fluent and well-written.

Reviewer 3 Report

1. I think Introduction is too poorly described. There are too few references to the literature to accurately present the new approach presented by the authors. It is necessary to indicate progress in a given topic on the basis of available publications. Indication of the advantages and disadvantages of existing solutions. Emphasize why the proposed solution is better/different than the others.

2. At the end of the Introduction, it is necessary to briefly present the content of the article.

3. Please read the two articles and refer to them in the Introduction:

https://doi.org/10.3390/en15010115

DOI:10.1016/j.epsr.2017.09.018

4. All variable subscripts should also be in italics.

5. Sentences with formulas do not end with commas or periods.

6. Why is the "M" in upright instead of italics? Is it a matrix? If so, it should be bolded.

7. Gridlines are needed in Figures.

8. There are no dots after chapters, but they happen in the article.

9. On what basis were the elements of the system and devices in the experiment so matched? what is the fault tolerance? what are the factors affecting the error of calculation results? How can bugs be eliminated?

10. Did the authors verify the results using a numerical method? For what reason has this not been considered / is it planned? What is the further direction of research?

11. Does the proposed method have the possibility of development? In what direction?

12. What are the significant results from these studies? Is it necessary to present the advantages and disadvantages of Conclusion on the basis of other solutions?

13. Literature formatting does not comply with the requirements of the journal. Are there older literature items, or is there no more recent research?

Extensive editing of English language required.